# The Use of Portable Devices for the Instrumental Assessment of Balance in Patients with Chronic Stroke: A Systematic Review

**DOI:** 10.3390/ijerph191710948

**Published:** 2022-09-02

**Authors:** Ana Mallo-López, Pilar Fernández-González, Patricia Sánchez-Herrera-Baeza, Alicia Cuesta-Gómez, Francisco Molina-Rueda, Ángela Aguilera-Rubio

**Affiliations:** 1International Doctorate School, Rey Juan Carlos University, 28933 Madrid, Spain; 2Department of Physiotherapy, Faculty of Sport Sciences, Universidad Europea de Madrid, Villaviciosa de Odón, 28670 Madrid, Spain; 3NeuroAvanza Neurological Physiotherapy Center, 28022 Madrid, Spain; 4Motion Analysis, Ergonomics, Biomechanics and Motor Control Laboratory (LAMBECOM), Department of Physical Therapy, Occupational Therapy, Rehabilitation and Physical Medicine, Faculty of Health Sciences, Rey Juan Carlos University, 28922 Madrid, Spain

**Keywords:** assessment, balance, devices, postural control, stroke, technology

## Abstract

Background: Improving balance remains a challenge in stroke rehabilitation. The technological development has allowed the design of more accessible and user-friendly systems for assessing postural control. Objectives: The aim of this review was to analyze portable devices for the instrumental assessment of balance in patients with chronic stroke. Methods: PRISMA guidelines were used to carry out the systematic review. The literature search was restricted to articles written in English or Spanish published from 2013 to December 2022 in Pubmed, Web of Science, Scopus, PEDro, and CINAHL. Of the 309 search results, 229 unique references were reviewed after duplicates were removed. The PEDro scale was applied to evaluate the methodological quality of the included papers, and the degree of evidence and level of recommendation were determined through the Oxford Centre for Evidence-Based Medicine. Results: A total of seven articles reporting on five different balance testing devices were included in this systematic review; they regarded BIORescue, a smartphone application, and the Biodex-BioSway Balance System for the evaluation of standing balance, and SwayStar Balance and Xsens ForceShoes™ for the evaluation of dynamic balance during walking. Conclusions: The use of portable devices that assess balance in adult patients with chronic stroke is scarce.

## 1. Introduction

Stroke causes a neurological deficit due to a focal injury with vascular origin in the central nervous system, whose etiology can be diverse. It is the disease that produces the greatest disability worldwide, and its economic cost has increased in recent decades [1,2].

One of the main causes of disability in patients who have suffered a stroke is an impaired balance. The lack of stability increases the risk of falls, restricts patients’ basic activities of daily life, and reduces their participation in the society [3]. Falls are one of the most common secondary complications of stroke; it was reported that 70% of stroke patients will suffer falls over the first year [4].

Postural control requires different sensory, neuromuscular, biomechanical, and cognitive components, involving complex processes of sensory integration and adapted motor response [5]. Due to the stroke, this complex multifactorial process is altered, making it extremely difficult to identify the main cause of the balance impairment. Improving balance remains a great challenge for neurorehabilitation [6,7].

Balance assessment instruments and scales are essential to evaluate the postural control. Traditionally, clinical scales have been used to identify balance deficits and the risk of falls [8]. These tools are quick and easy to use and do not require expensive materials. However, they offer a subjective evaluation and are less sensitive for the detection of changes in patients. In addition, these scales do not assess the systems involved in balance, the biomechanical aspects related to posture, or the compensatory strategies. In this sense, instrumental systems to measure postural control are more effective for balance evaluation [9,10].

Posturography is the more common instrumental method to assess balance. It uses systems that incorporate force platforms that provide quantitative data. In addition, posturography systems allow balance to be analyzed in different sensory conditions. This allows knowing the contribution of the visual, proprioceptive, and vestibular systems in postural control. Kinematic information can be offered by motion capture systems, accelerometers, or electrogoniometers [11].

Until recently, instrumental balance analysis systems were typically found only in research laboratories or large hospitals [12]. However, the technological development has allowed the design of low-cost systems that are more accessible and easier to handle. Force platforms, motion capture systems originally designed for videogames, wearables accelerometers, and virtual reality devices are some of them [13]. In addition, these instruments facilitate quality evaluations in the clinical setting, improving the connection between research and clinic application, which is one of the key points to the success of evidence-based clinical practice. Thus, these portable devices could be especially useful for patient’s associations and in daycare centers and small private neurorehabilitation centers where resources are limited and users are frequently chronic patients [14].

Previous reviews analyzed the psychometric properties and suitability of quantitative tools for the assessment of balance [15]. Research on assessment methods using smartphones and other technologies has shown good results [16]. However, qualitative studies have pointed out the potential difficulties of using these technologies in some populations [17]. Moreover, the background of physiotherapists and their previous ideas about technology have a significant influence on the choice of these technologies [18]. Scientific studies in chronic stroke rehabilitation are fewer than in acute or sub-acute rehabilitation phases, although more than half of stroke patients have a long-term disability [19]. To our knowledge, no previous studies analyzed the use of portable devices to assess balance in patients with chronic stroke.

Therefore, the aim of this systematic review was to analyze the portable devices for the instrumental assessment of balance in patients with chronic stroke.

## 2. Materials and Methods

### 2.1. Eligibility Criteria for Study Selection

We included studies meeting the following criteria: case series, observational studies, or experimental trials including participants older than 18 years of age diagnosed with chronic stroke, which used portable devices for the instrumental assessment of balance in these patients. According to Bernhardt et al., the chronic phase of stroke starts 6 months after the injury [20].

Exclusion criteria were as follows: (1) studies including participants with acute or subacute stroke; (2) studies using portable devices for the treatment of balance in stroke patients.

### 2.2. Search Strategy for the Identification of the Studies

The search for articles was carried out from May to June 2022. The databases consulted were: Pubmed, PEDro, Web of Science (WoS), Scopus, and CINAHL. The publication date was set from 2013 to 2022. Only scientific articles published in English and Spanish were included.

The search strategy was carried out with the following keywords: “balance”, “postural control”, “upright position”, “stance”, “assessment”, “stroke”, “instrument”, “technology”, “devices”, combining them by means of the Boolean operators AND and OR in the different searches.

### 2.3. Review Methods

A selection of titles and abstracts of the results found in the databases was performed. We assessed the content of the selected and identified studies that met the inclusion criteria, jointly established by two authors (A.M.-L. and P.F.-G.). Subsequently, two authors (A.M.-L. and P.S.-H.-B) independently read the full texts of the selected articles that met the inclusion criteria and ranked them according to their relevance. In order to improve the quality of the present systematic review, the guidelines of the PRISMA statement [21] were followed.

### 2.4. Risk of Bias

Each article in this systematic review was carefully examined with the Cochrane risk of bias tool. The following biases were considered and critically analyzed: selection bias, performance bias, detection bias, attrition bias, and between-study reporting bias. These details are mentioned in the results section.

### 2.5. Methodological Quality Assessment

The methodological quality of the selected papers was assessed using the PEDro scale [22]. It was decided to evaluate the clinical utility of each measurement tool based on the studies by Tyson and Connell [15]. This tool consists of 4 items: time required to complete the test (scores: 3 = <10 min; 2 = 10–30 min; 1 = 30–60 min; 0 = >1 h); cost of the equipment (3 = <$188; 2 = $188–$938; 1 = $938–$1,875; 0 = >$1,875; training and equipment (2 = no; 1 = yes, but simple and clinically feasible; 0 = yes, but not clinically feasible or unknown); portable tool: “Can it be carried by the patient?” (2 = yes, easily; 1 = yes, in a case or trolley; 0 = no or very difficult). The maximum score is 10 points, and according to the authors, scores between 9 and 10 indicate that the measurement tool can be recommended for clinical use.

In addition, the Oxford scale [23] was applied to all selected articles to determine the levels of evidence and recommendation, rating the level of evidence according to the best design for each clinical scenario.

## 3. Results

We found 309 studies in the databases. A total of 80 papers were discarded as duplicates, leaving 229 to be analyzed. After a title and abstract review, 43 studies remained. Finally, a total of seven articles [24,25,26,27,28,29,30] were included in the present systematic review. Thus, 36 works were excluded, as they did not meet the inclusion criteria (Figure 1).

All included studies enrolled 159 people with chronic stroke (97 were men, and 62 were women; 81 had right-sided hemiplegia, 76 had left-sided hemiplegia, and 2 had both-sided hemiplegia). Several studies specified that they included patients with ischemic or hemorrhagic strokes [24,25,26].

A summary of the characteristics of the included studies [24,25,26,27,28,29,30] is presented in Table 1. The main results of each paper are presented according to the following criteria: year of publication (in descending chronological order), type of study, sample characteristics, outcome measures, and type of device.

### 3.1. Summary of the Main Results

The studies included in this review used the following devices to analyze the balance or the postural control of the subjects with chronic stroke: BIORescue [24,25,26], SwayStar Balance [27], Biodex-BioSway Balance System [28], Xsens ForceShoes™ [29], and a smartphone application [30].

These devices allow obtaining various outcome measures. BIORescue can provide the length [24,25,26] and the velocity [25,26] of the center of pressure (COP), as well as the limit of stability (LOS) [25]. SwayStar Balance and the smartphone application allow data to be collected on the trunk angular displacement [27,30]. Xsens ForceShoes™ provides reference data to the COP and the center of mass (COM) [29]. Finally, the Biodex-BioSway Balance System provides data regarding the Postural Stability Test (PST), the Limits of Stability Test (LOS), and the Modified Sensory Organization Test (MSOT) [28].

Most of the devices included in this review assess static balance. For this, the subjects maintain the standing position during the measurement [24,25,26,28,30]. Maguire et al. (2020) and van Meulen et al. (2016) obtained postural control data while the subjects were walking with the devices used [27,29].

All studies except one [30] used clinical scales to assess patients’ balance in addition to the devices:Two studies (60 participants) used the Postural Assessment Scale for Stroke (PASS) [25,26].Three studies (87 participants) used the Berg Balance Test (BBT) [26,28,29].Three studies (84 participants) used the Time Up and Go test (TUG) [24].One study (4 participants) used the Functional Gait Assessment (FGA) [27].One study (50 participants) used the BESTest [28].One study (50 participants) used the Activities Specific Balance Confidence scale (ABC) [28].

The intervention was carried out in university settings in three studies [25,26,30], in rehabilitation hospitals and/or clinics in other three studies [24,27,29], and was not specified in one study [28].

### 3.2. Methodological Quality

The PEDro scale was used. The following scores were obtained: 9 [24], 8 [25,26], 5 [27], and 4 [28,29,30] in the included studies (Table 2).

Regarding to the clinical utility of each measurement tool, the results showed that the Biodex-BioSway Balance System scored 5 points, the BIORescue scored 6 points, the SwayStar Balance System and the Xsens ForceShoes TM scored 7 points, and the system that uses a smartphone scored 9 points (Table 3).

The Oxford scale was used to assess both the levels of evidence and the recommendation of the included articles. A level 1b was obtained in one of the studies [25], a level 2b in three studies [24,26,28], a level 3b in one of them [30], and a level 4 in the remaining studies [27,29] (Table 4).

### 3.3. Risk of Bias within Studies

The results of the bias analysis are presented in Figure 2.

## 4. Discussion

The purpose of this review was to evaluate, by compiling and critically reading the published literature, the use of portable devices for the instrumental assessment of balance in patients with chronic stroke, identifying the most frequently used devices and their main characteristics.

We believe that the search strategy was detailed and selected the most relevant papers.

In the seven articles reviewed, five different portable devices were used for balance assessment. Five of them studied [24,25,26,28,30] standing balance, and only two studied [27,29] dynamic balance during walking. Considering that 20% of stroke patients are unable to walk independently [31], it is not surprising that most of the articles studied standing balance, although it is remarkable that none of the articles assessed balance in other postural sets, such as sitting, stepping, or single-leg standing.

In the evaluation of the clinical usefulness of each device, the cost, difficulty of assembly, and user-friendliness were taken into account, ignoring the economic costs of the data processing software and the subsequent analysis. Likewise, the time required for the analysis of the different aspects of equilibrium was short, thus facilitating the evaluations. Regarding the portability, except for the Biodex-BioSway Balance System [28], which is bulky, the remaining devices are easily portable by patients [24,25,26,27,29,30].

As described in the results, the devices were used in laboratories, hospitals, and clinics. The use of portable systems would be especially useful in small centers or patient associations where chronic users tend to predominate. However, this is not the preferred place of use. Implementing such quantitative assessment methods would be a major boost to link clinical care and research, improving patient interventions and providing reliable and validated knowledge to the scientific community.

### 4.1. Standing Balance

The most frequent evaluation method chosen by the researchers was posturography, through BioRescue RM and Biodex-BioSway Balance System.

The first device appears in three of the reviewed articles published by the same author [24,25,26], the methodology and type of study being similar. These three studies did not use the same outcome measures; the study published in 2019 only considered the COP length trajectory with eyes closed and open, while the study published in 2021 also analyzed the mean COP velocity under the same conditions. In addition to these parameters, the study published in 2020 included the LOS.

Sahin et al. [28] chose the Biodex-BioSway Balance System to perform the MSOT. This test also allows the assessment of the mean speed and path length of COP with closed and open eyes on a stable and unstable platform. Only the final test score was considered in the study. In addition, the Postural Set Test and the LOS were recorded while the participants were standing on both legs.

One study used a mobile application to assess standing balance. The mobile was secured with a belt to the patient’s waist and, through an Android^®^ application, recorded changes in trunk tilt angles. These data issued by the mobile application were processed with a software, obtaining the trunk tilt projection that allowed estimating the mean velocity and its trajectories in the mediolateral and anteroposterior axes. The stroke participants remained in Romberg position with eyes open, while more challenging positions were attained only by the healthy subjects.

Therefore, the outcome measures obtained by the different systems are very similar when assessing standing balance, focusing on the displacement and velocity of the center of pressures or trunk under different conditions for static balance and on the LOS test for dynamic balance. All measurements were performed while the stroke patients were standing with both feet supported and on a stable surface, except in Sahin et al., which also performed measurements on an unstable surface, as required by the MSOT.

All three devices require a computer with corresponding software. In the case of the mobile application, a Wi-Fi connection is also required.

In relation to the psychometric properties of the portable systems, the use of validated instrumental analyses that provide robustness and reliability to the data is essential. In the case of these three systems, to our knowledge, one study analyzed their validity in relation to a goal standard [32]. This study indicated acceptable reliability for BioSway™ and NeuroCom^®^ SMART EquiTest System, with better agreement when the assessment of postural sway was performed on stable/static surfaces. However, the authors studied a healthy population, and some of the tests performed involved head movement. These devices are not currently validated for the three tests performed in the Sahin et al. [28] study, nor for subjects with stroke.

In the case of BioRescue, to our knowledge, there is only one small study whose aim was to demonstrate the reproducibility of the limits of the stability test in healthy subjects. Therefore, we do not know whether the data obtained with the device are comparable with data from laboratory posturography [33].

The use of mobile applications is becoming increasingly widespread due to the easy handling, portability, and accessibility of these apps. A recent systematic review reported preliminary evidence supporting that smartphone-based gait and balance assessments are valid, reliable, sensitive, and specific in laboratory settings in stroke patients [16]. In fact, a specific application has been developed to assess balance in stroke patients and has been confirmed to be a convenient, reliable, and valid tool for the assessment of balance in this population [34]. However, we did not find any study that used this application to assess the balance of stroke participants.

Similarly, it is striking that existing posturography systems for assessing standing balance in stroke patients, which are inexpensive, portable, and easy to use, are not being used in research studies. Llorens et al. designed the “Posturography Test” which uses the Wii Board Balance as a force platform and allows performing the modified Clinical Test of Sensory Interaction on Balance (mCTSIB), the Limits of Stability (LOS) test, and the Rhythmic Weight Shift (RWS) test, presenting excellent psychometric properties and being a valid, reliable, and feasible tool to identify the balance performance of people with stroke compared to healthy subjects [35]. The software to perform the tests is free and open-access and requires Bluetooth and no internet connection, and the Wii Board Balance is a video game accessory whose price is more than affordable. None of the articles that appeared in the search used this device, even though it was specifically validated for stroke survivals.

### 4.2. Dynamic Balance during Walking

Two different systems based on the use of inertial sensors were employed to assess dynamic balance during walking: the Sway Star Balance System and Xsens ForceShoes™.

The first is a device that is placed on the subject at the lumbar level, where the COM reference is located. The device’s two angular velocity sensors capture the displacement of the COM in the sagittal and frontal planes during walking, obtaining the Trunk Sway. Through Bluetooth, the data are processed on a PC calculating the Total Angle Area (TAA) in degrees. The Sway Star Balance System has been shown to be comparable to dynamic computerized posturography in healthy subjects. However, we do not know its reliability in chronic stroke patients [36].

Xsens ForceShoes™ incorporate an inertial measurement unit (IMU), a 3D force/moment sensor, and ultrasounds. Using MATLAB software, the COM position, base of support, and dynamic stability margins can be estimated. In addition, the step length, velocity, and time in monopodal support of each foot are compared. This system makes it possible to obtain objective data on the subject’s balance and gait ability in real-life patient environments. In 2009, a study concluded that the instrumented shoes allow for an accurate and continuous estimation of COM displacement [37].

Both devices require a computer with data processing software, that used in the Xsens ForceShoes™ being much more complex.

To carry out a more complete evaluation of stroke subjects, it is very useful to combine balance analysis systems with motion capture systems. In recent years, new low-cost methods of motion analysis have been developed [38], which makes it possible to obtain objective data in clinical settings. These are based on the use of videos captured by video cameras or mobile smart devices for further analysis. There are different applications for mobile phones or software for computers, such as Kinovea^®^ [38,39]. In 2020, a study evaluated the reliability and the agreement of this software with a three-dimensional motion capture system for gait analysis in healthy subjects [40]. However, its psychometric properties have not been studied in stroke patients. Another tool used in the analysis of movement is the Kinect^®^ sensor of the Xbox 360 video game console, with which a three-dimensional analysis is obtained. The Kinect^®^ has advantages in terms of cost and portability, being an effective clinical tool to obtain kinematic and spatiotemporal gait parameters [41,42]. The literature reports a study of its validity and reliability in subjects with stroke [43].

Although there are portable devices validated for balance assessment in patients with chronic stroke, they are not being used, and it would be advisable to disseminate and facilitate their use, particularly in patients living with a long-term disability, whose access to research laboratories is reduced. Closer links between research and clinical practice are one of the foundations of evidence-based clinical practice, and these types of devices may help in this regard [14].

Further investigations studying the possible causes of the limited diffusion of these devices, which limits the methodological quality of research and the development of better therapeutic approaches, are needed; these causes may include the inappropriate researcher’s perceptions of these devices or the unsuccessful dissemination of evidence-based knowledge.

This systematic review has certain limitations that need to be mentioned, mainly due to the significant clinical diversity of the included studies and their methodological limitations. Firstly, the small number of publications on this topic is noteworthy, indicating the need for studies of adequate methodological quality. The heterogeneity of the different devices, the intervention protocols, and the outcome measures used should be highlighted. Another limitation is the difficulty in accessing information on the technical characteristics of each device. Finally, this systematic review only selected studies published in English and Spanish.

## 5. Conclusions

In the last 10 years, despite advances in technology, the use of portable devices that assess balance in adult patients with chronic stroke has been limited. Five devices have been used: BioRescue, the Biodex Bio-Sway Balance System, and an Android smartphone app to evaluate standing balance, the Sway Star Balance System and Xsens ForceShoes™ to evaluate dynamic balance during walking. Except for the instrumented shoes, none of the devices have been shown to be suitable for a quantitative balance assessment in stroke patients.

## Figures and Tables

**Figure 1 ijerph-19-10948-f001:**
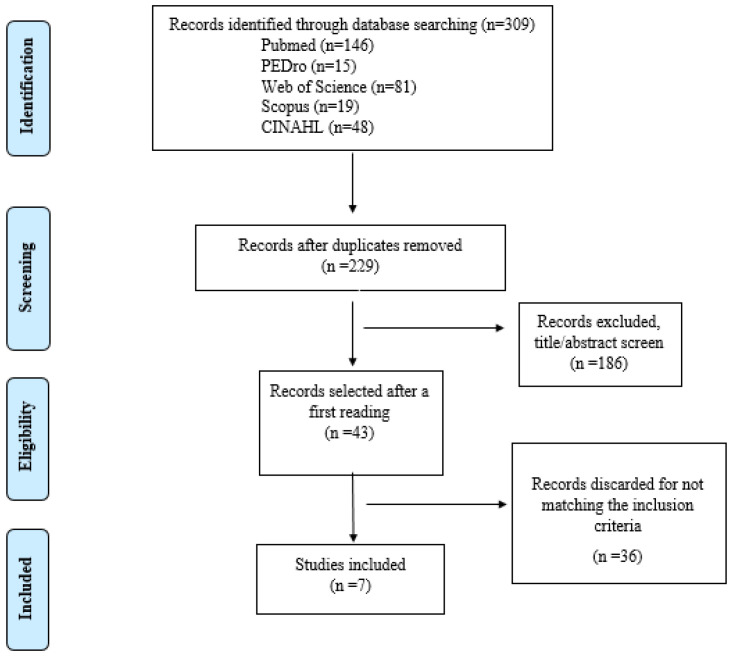
Flow chart.

**Figure 2 ijerph-19-10948-f002:**
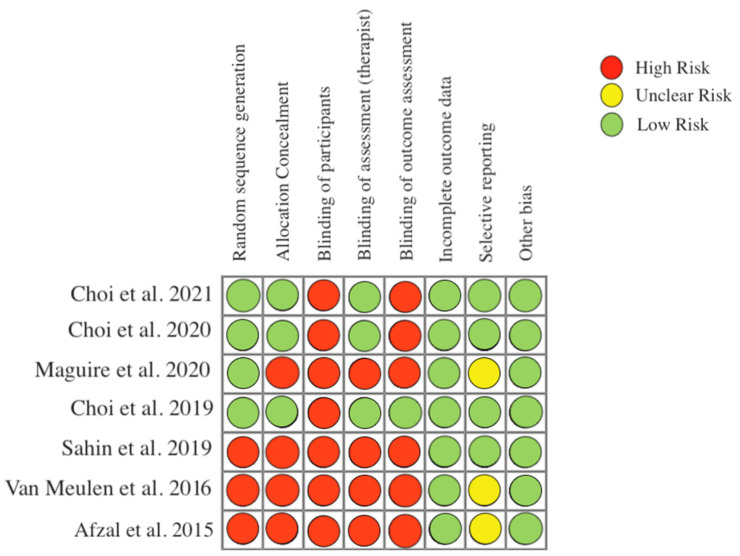
Risk of bias [24,25,26,27,28,29,30].

**Table 1 ijerph-19-10948-t001:** Summary of the results.

Author	Type of Study	Sample Characteristics	Outcome Measures	Type of Device
Choi et al. (2021) [26]	RCT	Chronic stroke (N = 24) Age (years): CG 67.4 ± 12.9; IG 64.1 ± 10.5 Sex: CG 7M; 5F; IG 7M; 5F Paretic side: CG 7 right; 5 left; IG 6 right; 6 left Stroke type: CG 3 hemorrhage; 9 infarction IG 1 hemorrhage; 11 infarction Time since stroke (months): CG 9.8 (8.0); IG 9.5 (8.4) Height (cm): CG 161.6 ± 8.8; IG 163.5 ± 8.2 Weight (kg): CG 60.8 ± 9.4; IG 59.8 ± 8.9 Trunk impairment scale (score): CG 12.1 (1.9); IG 12.8 (2.1)	BBT TUG PASS Length and velocity of the COP	BIORescue (analysis system by biofeedback, RM INGENIERIE, Rodez, France)
Choi et al. (2020) [25]	RCT	Chronic stroke (N = 36) Age (years): CG 67.4 ± 12.9; PG 64.1 ± 10.5 NPG 62.4 ± 12.1 Sex: CG 7M; 5F; PG 7M; 5F; NPG 6M; 6F Paretic side: CG 3 right; 9 left; PG 6 right; 6 left NPG 7 right; 5 left Stroke type: CG 5 hemorrhage; 7 infarction PG 6 hemorrhage; 6 infarction NPG 4 hemorrhage; 8 infarction Time since stroke (months): CG 9.7 (8.0); PG 9.5 (8.4); NPG 9.4 (3.8) Height (cm): CG 161.6 ± 8.8; PG 163.5 ± 8.2 NPG 164.1 (10.1) Weight (kgs): CG 60.8 ± 9.4; PG 59.8 ± 8.9; NPG 62.8 (12.0)	PASS TUG COL path length COL path speed LOS	BIORescue (analysis system by biofeedback, RM INGENIERIE, Rodez, France)
Maguire et al. (2020) [27]	CT	Chronic stroke (N = 4) Age (years): 50–58 Sex: 3M; 1F Paretic side: 3 right; 1 left Time since stroke (years): 4.5 (± 1.6) Height (m): 1.72–1.88 Mini Mental State Score > 22 BBT ≥ 43	FGA Trunk angular displacement	SwayStar Balance System: two angular velocity sensors (Fibreoptic gyroscopes) which are attached to a belt and worn by the participants at the level of L2/3 (CoM)
Choi et al. (2019) [24]	RCT	Chronic stroke (N = 24) Age (years): CG 59.7 ± 10.2; IG 62.8 ± 4.8 Sex: CG 8M; 4F; IG 8M; 4F Paretic side: CG 5 right; 7 left; IG 7 right; 5 left Stroke type: CG 7 hemorrhage; 5 infarction IG 5 hemorrhage; 7 infarction Time since stroke (months): CG 73.0 (31.9); IG 67.2 (43.8) Height (cm): CG 162.9 ± 8.6; IG 166.0 ± 9.4 Weight (kg): CG 63.4 ± 10.5; IG 67.8 ± 8.5	TUG Static balance ability: COP path length	BIORescue (analysis system by biofeedback, RM INGENIERIE, Rodez, France)
Sahin et al. (2019) [28]	CT	Chronic stroke (N = 50) Age (years): Faller 53.33 ± 18.93 Non-Faller 64.03 ± 14.65 Sex: Faller 12M; 14F Non-Faller 18M; 6F Paretic side: Faller 15 right; 11 left Non-faller 17 right; 7 left Dominant side: Faller, 26 right; 0 left Non-Faller 22 right; 2 left Time since stroke (months): Faller 30.00 (6.00–60.00) * Non-Faller 33.00 (9.00–89.25) * Modified Rankin Scale (0–6 point): Faller 3 (1–4) * Non-Faller 1 (0–2) *	BESTest BBT ABC PST LOS MSOT	Biodex-BioSway Balance System (SD 950-340, Biodex Medical Systems, Inc., Shirley, NY, USA)
van Meulen et al. (2016) [29]	CT	Chronic stroke (N = 13) Age (years): 64.1 ± 8.7 Sex: 8M; 5F Paretic side: 2 right; 11 left Dominant side: 12 right; 1 left Time since stroke (years): 2.4 ± 1.8 Height (cm) 173 ± 9.74 Weight (kg) 87 ± 10.89 BBT ≥ 35 Walking aid (N = 5): 3 St; 1 AFO; 2 OS	BBT COP COM	Xsens ForceShoes™ (Xsens Technologies B.V., Enschede, The Netherlands) additionally equipped with ultrasound sensors (instrumented shoes)
Afzal et al. (2015) [30]	CT	Chronic stroke (N = 8) Age (years): 52 ± 11.9 Sex: 6M; 2F Paretic side: 3 right; 3 left; 2 bilateral Height (cm) 169 ± 6.3 Weight (kgs) 62 ± 5.8 Mini-Mental State Examination mean 21.5	Trunk tilt angles	Smartphone application

* Median (25–75 IQR, interquartile range); CT: Clinical trial; RCT: Randomized clinical trial; CG: Control group; IG: Intervention group; PG: Paretic group; NPG: Non-paretic group; St: Stick; AFO: Ankle foot orthosis; OS: Orthopedic shoes; M: Male; F: Female; PASS: Postural Assessment Scale for Stroke; BBT: Berg Balance Test; TUG: Time Up and Go test; COP: Center of pressure; COL: Center of loading; LOS: Limit of stability; FGA: Functional Gait Assessment; ABC: Activities-Specific Balance Confidence Scale; PST: Postural Stability Test; MSOT: Modified Sensory Organization Test; COM: Center of mass; TB: Tinetti Balance test; FM: Fugl–Meyer.

**Table 2 ijerph-19-10948-t002:** PEDro scale.

Ítem	1	2	3	4	5	6	7	8	9	10	11	Total
Author-Year												
Choi et al. (2021) [26]	-	1	1	1	0	0	1	1	1	1	1	8/11
Choi et al. (2020) [25]	-	1	1	1	0	0	1	1	1	1	1	8/11
Maguire et al. (2020) [27]	-	1	0	0	0	0	0	1	1	1	1	5/11
Choi et al. (2019) [24]	-	1	1	1	0	1	1	1	1	1	1	9/11
Sahin et al. (2019) [28]	-	0	0	0	0	0	0	1	1	1	1	4/11
Van Meulen et al. (2016) [29]	-	0	0	0	0	0	0	1	1	1	1	4/11
Afzal et al. (2015) [30]	-	0	0	0	0	0	0	1	1	1	1	4/11

Scoring: 0 = the assessment criterion is not achieved; 1 = the evaluation criterion is achieved.

**Table 3 ijerph-19-10948-t003:** Clinical utility of the selected measurement tools.

Measurement Tool	Time	Cost	Specialist Equipment	Portability	Total (Max = 10)
BIORescue	3	0	1	2	6
SwayStar Balance System	3	0	2	2	7
Biodex-BioSway Balance System	3	0	1	1	5
Xsens ForceShoes^TM^	3	0	2	2	7
Smarthphone	3	3	2	2	10

**Table 4 ijerph-19-10948-t004:** Oxford Scale.

Article	Level of Evidence	Level of Recommendation
Choi et al. (2021) [26]	2b	B
Choi et al. (2020) [25]	1b	A
Maguire et al. (2020) [27]	4	C
Choi et al. (2019) [24]	2b	B
Sahin et al. (2019) [28]	2b	B
Van Meulen et al. (2016) [29]	4	C
Afzal et al. (2015) [30]	3b	B

## Data Availability

Data are available in the manuscript.

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
