# Peer review of "The Use of Portable Devices for the Instrumental Assessment of Balance in Patients with Chronic Stroke: A Systematic Review"

_ijerph, 2022, doi:10.3390/ijerph191710948_

Round 1
Reviewer 1 Report
Please refer to the attachment thank you for your help.

Author Response
1.Introduction: Please describe study design for the review in the context?Please describe the rationale for the review in the context of what is already known?
Thank you for the comment. We have specifically studied portable devices used in chronic stroke. We have added in line 71 two current reviews with similar topics but they analyse different devices, portable or not, in general stroke population. We have added in line 76 the reasons for the need for this review.
2.Study selection: What are the conditions for chronic stroke screening?
Thank you very much for your comment. We have added in line 88 the conditions for chronic stroke.
3. Results: What is the reason for the deletion of 80 duplicate literature?
Thank you for the comment. After reviewing various databases (Pubmed, PEDro, Web of Science (WoS), Scopus and CINAHL), we found 309 studies. We discarded 80 studies as they were duplicates appearing in at least two of these databases.
4.Please indicate the number of 7 references (p4-p6 and p7).
Thank you very much for the comment; we have added the number of references in the indicated pages.
5.What are the easy to carry balancing devices for stroke patients in the 7 literature? [18-21,23,24]
Thank you for the comment. Table 1 shows each of the devices associated with these references and according to the scale used to assess clinical utility (specified on p.3, line 122), devices are categorized as easily portable, portable as they are similar to a suitcase, or non-portable.
6.How is this manuscript applied clinically? What is the contribution?
Thank you for the suggestion. In p. 9-11 we describe the different devices and we mention free devices that are validated which use in clinical setting could be useful. In line 336 we support the use of objective assessments in clinical settings to improve interventions.
7.Please provide a general interpretation of the results in the context of other evidence, and implications for future research?
Thank you for your comment. We have added the implications for future research in line 342.
8.What are the restrictions on data collection for your manuscript?
Thank you for the comment. Specified on p.12, line 351 we have mentioned difficulties for accessing to devices technical information and costs.
9.Suggestions for writing this manuscript?
Thank you for your comment. On our day to day work we have found lots of studies related to quantitative balance assessment but there is a lack of documentation for the specific subset of portable devices applied to chronic stroke. This analysis will enrich the existing knowledge to develop long-term studies with larger sample sizes, as research methodology requires, to improve the interventions for patients with chronic stroke.
We would like to thank the reviewer for taking the time to thoroughly review our manuscript and provide recommendations for improving it.
Reviewer 2 Report
The aim of this review is to evaluate the portable devices in patients with chronic stroke. However, there are couple similar reviews in the literature for the same purpose. Thus the author need specify what is the difference between their review and others.
Author Response
The aim of this review is to evaluate the portable devices in patients with chronic stroke. However, there are couple similar reviews in the literature for the same purpose. Thus the author need specify what is the difference between their review and others.
Thank you for your comment. We have specified in lines 71 and 285, the differences between our review and others.
We would like to thank the reviewer for taking the time to thoroughly review our manuscript and provide recommendations for improving it.
Reviewer 3 Report
The manuscript is well written and organized. However, the scientific contribution and importance of this work are not highlighted. In addition to that, future directions could be added.
You could add additional information about the inclusion criteria in the Review methods section.
Author Response
The manuscript is well written and organized. However, the scientific contribution and importance of this work are not highlighted. In addition to that, future directions could be added.
Thank you for your comment. We have added the information required in p. 12, line 342.
You could add additional information about the inclusion criteria in the Review methods section.
Thank you very much for your suggestion. In material and methods, we have added additional information about the inclusion criteria.
We would like to thank the reviewer for taking the time to thoroughly review our manuscript and provide recommendations for improving it.
Round 2
Reviewer 1 Report
Dear Authors:
Review manuscripts are accepted in present form.
Blessings of peace and health
Reviewer 2 Report
I have no questions with current version.